# Mechanical Disturbance of Osteoclasts Induces ATP Release That Leads to Protein Synthesis in Skeletal Muscle through an Akt-mTOR Signaling Pathway

**DOI:** 10.3390/ijms23169444

**Published:** 2022-08-21

**Authors:** Camilo Morales-Jiménez, Julián Balanta-Melo, Manuel Arias-Calderón, Nadia Hernández, Fernán Gómez-Valenzuela, Alejandro Escobar, Enrique Jaimovich, Sonja Buvinic

**Affiliations:** 1Institute for Research in Dental Sciences, Faculty of Dentistry, Universidad de Chile, Santiago 8380544, Chile; 2Department of Basic Sciences of Health, Faculty of Health Sciences, Pontificia Universidad Javeriana, Cali 760031, Colombia; 3School of Dentistry, Faculty of Health, Universidad del Valle, Cali 760043, Colombia; 4Institute of Biomedical Sciences, Faculty of Medicine, Universidad de Chile, Santiago 8380453, Chile; 5Center for Exercise, Metabolism and Cancer Studies CEMC2016, Faculty of Medicine, Universidad de Chile, Santiago 8380453, Chile

**Keywords:** bone-muscle interactions, muscle physiology, bone remodeling, osteoclasts, purinergic signaling, extracellular ATP

## Abstract

Muscle and bone are tightly integrated through mechanical and biochemical signals. Osteoclasts are cells mostly related to pathological bone loss; however, they also start physiological bone remodeling. Therefore, osteoclast signals released during bone remodeling could improve both bone and skeletal muscle mass. Extracellular ATP is an autocrine/paracrine signaling molecule released by bone and muscle cells. Then, in the present work, it was hypothesized that ATP is a paracrine mediator released by osteoclasts and leads to skeletal muscle protein synthesis. RAW264.7-derived osteoclasts were co-cultured in Transwell^®^ chambers with flexor digitorum brevis (FDB) muscle isolated from adult BalbC mice. The osteoclasts at the upper chamber were mechanically stimulated by controlled culture medium perturbation, resulting in a two-fold increase in protein synthesis in FDB muscle at the lower chamber. Osteoclasts released ATP to the extracellular medium in response to mechanical stimulation, proportional to the magnitude of the stimulus and partly dependent on the P2X_7_ receptor. On the other hand, exogenous ATP promoted Akt phosphorylation (S473) in isolated FDB muscle in a time- and concentration-dependent manner. ATP also induced phosphorylation of proteins downstream Akt: mTOR (S2448), p70S6K (T389) and 4E-BP1 (T37/46). Exogenous ATP increased the protein synthesis rate in FDB muscle 2.2-fold; this effect was blocked by Suramin (general P2X/P2Y antagonist), LY294002 (phosphatidylinositol 3 kinase inhibitor) and Rapamycin (mTOR inhibitor). These blockers, as well as apyrase (ATP metabolizing enzyme), also abolished the induction of FDB protein synthesis evoked by mechanical stimulation of osteoclasts in the co-culture model. Therefore, the present findings suggest that mechanically stimulated osteoclasts release ATP, leading to protein synthesis in isolated FDB muscle, by activating the P2-PI3K-Akt-mTOR pathway. These results open a new area for research and clinical interest in bone-to-muscle crosstalk in adaptive processes related to muscle use/disuse or in musculoskeletal pathologies.

## 1. Introduction

The musculoskeletal system changes through ontogeny. Several clinical trials show the simultaneous alteration of both muscle mass and bone mass during the life cycle of humans [1,2,3]. The lack of physical activity during conditions such as cardiac failure, neuromuscular lesions, rigid fracture reductions, microgravity, and prolonged rest results in a significant loss of muscle and bone tissues, whereas mechanical stimulation and physical exercise produce the opposite effect [4,5,6,7,8,9,10,11,12]. Thus, the muscle–bone interaction could be explained from a mechano-functional hypothesis. However, the biomechanical interaction does not entirely support the regulation of the musculoskeletal system during activities like swimming and cycling, in which the muscle mass increases without a proportional rise in bone mass [6].

Another potential regulatory mechanism between muscle and bone tissues is molecular communication. A preclinical study in rats determined that the quality of the newly formed mineralized tissue after bone fracture reduction was dependent on the pore size of a membrane inserted locally between muscle and bone, without affecting the mechanical interaction through the tendon [13]. These findings suggest that different soluble factors play a role in the molecular communication between muscle and bone in physiological and pathological scenarios. In addition, both muscle and bone are considered secretory organs [14,15]. The cells responsible for the bone-remodeling osteoblasts (bone apposition cells), osteocytes (bone mechanosensory cells), and osteoclasts (bone resorption cells) [16] differentially release soluble factors to the extracellular milieu that may impact the structure and function of the skeletal muscle, such as Dickkopf-related protein 1 (DKK-1), FGF-23, matrix extracellular phosphoglycoprotein (MEPE), osteoprotegerin (OPG), sclerostin, RANKL (Receptor activator of nuclear factor kappa-Β ligand) and ATP [4,7,14,16].

Due to their bone resorption capability, the osteoclasts are mostly related to bone loss. However, it is relevant to note that the osteoclast activity starts all the bone remodeling processes in physiological conditions. Bone remodeling involves the local degradation of bone tissue before the new bone formation [17]. Then, since the osteoclasts are secretory cells and, in addition, the resorption process adds free molecules to the local environment [17], it may be considered that the resorptive activity could be related to a molecular communication with the surrounding tissues like the skeletal muscle. Therefore, osteoclast-derived signals released during bone remodeling could improve both bone and skeletal muscle mass.

A signaling pathway shared by bone and muscle for their proper function is that activated by extracellular nucleotides (ATP, UTP and its metabolites) [18]. The signaling pathways mediated by extracellular ATP (eATP) are some of the most primitive and ubiquitous mechanisms for cell-to-cell interaction. ATP and UTP are released by cells, either at rest or after stimuli such as shear stress, hypoxia, or inflammation. ATP signalize through activation of P2X/P2Y receptors located at the plasma membrane. P2X_1–7_ are ion channels non-selective for Na^+^/Ca^2+^ and P2Y_1,2,4,6,11,12–14_ are metabotropic receptors coupled to trimeric G-proteins [19]. The purinergic signaling has been widely described in the musculoskeletal system. P2X/P2Y receptors are expressed in all the bone-remodeling cells as well as in skeletal muscle fibers [18,20]. Both osteoblasts and osteocytes release ATP at rest and after mechanical stimulation [21,22]. eATP has been related to the bone anabolic response to mechanical loading and to the bone repair process after micro-damages [23]. eATP evokes osteoblast proliferation, differentiation, and bone mineralization [24,25,26]. In osteocytes, eATP controls mechanotransduction and is used for communicating with neighbor osteocytes or osteoblasts [21]. In osteoclasts, the role of extracellular nucleotides is controversial. While some evidence showed that uridine nucleotides promote osteoclast differentiation and bone resorption via P2Y_6_ receptor activation [27,28], other findings demonstrated that ATP basally released by osteoclasts evokes cytoskeletal disorganization that impairs bone resorption and promotes osteoclast death [29], similar to the effect of mechanical stimuli [30,31]. eATP release from osteoclast has been only described at rest, but no studies under mechanical stimulation have been addressed [32].

Extracellular nucleotides also play a central role in skeletal muscle physiology in health and disease, as has been demonstrated by us and others [18,20]. P2Y and P2X receptors are widely expressed in skeletal muscles and are involved in structuration and maintenance of the neuromuscular junction [33,34]. P2Y receptors promote myoblasts proliferation and differentiation and protect myotubes against apoptosis [35,36,37]. We have demonstrated that ATP is a relevant mediator between membrane depolarization, calcium signaling, and gene expression, both in skeletal primary cultures and adult skeletal fibers [38,39,40]. Our studies place extracellular ATP as a relevant mediator of the excitation–transcription coupling, through regulation of muscle gene synthesis and plasticity [20]. eATP released by depolarization also mediates glucose uptake by activating the PI3Kγ–Akt-AS160 pathway that leads GLUT4 translocation to the cell surface [41]. Recently, Ito et al. have demonstrated that eATP promotes skeletal muscle hypertrophy mediated by the mTORC1 pathway [42]. Moreover, the regulation of protein synthesis due to the activation of PI3K-Akt-mTOR in the presence of eATP has been demonstrated in fibroblasts and neurons [43,44]. Therefore, in this work it was hypothesized that the eATP released from bone cells after mechanical stimulation increases the protein synthesis in skeletal muscle through the Akt-mTOR signaling pathway. Bearing in mind that the ATP release from osteocytes and osteoblasts after mechanical stimulation has been previously established, this research focused on the ATP release from osteoclasts and its relationship with protein synthesis in skeletal muscle. It is proposed that eATP could be a master regulator of bone-muscle crosstalk.

## 2. Results

### 2.1. Mechanically Stimulated Osteoclasts Lead to Protein Synthesis in Co-Cultured FDB Muscle

We developed a model of indirect co-culture between RAW-derived osteoclasts and mouse FDB isolated muscle using Transwell^®^ chambers, with a 0.4 µm pore-size membrane (Figure 1A). The isolated muscle was placed and stabilized in the lower compartment, and then an upper chamber with previously purified osteoclasts was positioned. Osteoclasts were mechanically stimulated by pipetting the culture medium, and global protein synthesis was addressed in the FDB muscle by the SUnSET pulse-chase approach, as detailed in Methods. When pipetting was performed without osteoclasts in the upper chamber, no changes were observed in puromycin staining in muscle lysates, discarding a putative effect of medium perturbing. However, after mechanical stimulation of osteoclasts in the upper compartment, a two-fold increase in puromycin staining was observed in muscle lysates, suggesting an improved rate of protein synthesis (Figure 1B). As a control of a described trophic signal, the effect of 100 nM insulin was addressed on FDB isolated muscles. Insulin evoked a 1.7-fold increase in the puromycin staining of muscle lysates (Figure 1C). These results suggest that mechanically stimulated osteoclasts release a soluble factor able to increase protein synthesis in the whole FDB muscle.

### 2.2. RAW 264.7 Release ATP by Mechanical Stimulation

When looking for a putative mediator for osteoclast to muscle crosstalk, extracellular ATP was considered. Then, it was addressed if mechanical stimulation of osteoclasts evoked ATP release. We worked with RAW 264.7 cells differentiated to osteoclasts through 5d-incubation with RANKL. In these conditions, the TRAP-positive and multinucleated phenotype characteristic of osteoclasts was observed (Figure 2A), as well as an increase in the mRNA for the osteoclast markers TRAP, cathepsin K, metalloproteinase 9, carbonic anhydrase, lysosomal ATPase and integrin β1 (Figure 2B). In these cells, the mechanical stimulation by pipetting increasing volumes of extracellular medium (0–12–15–37–50% of total medium) evoked a rapid release of ATP at 15 s, with half-life times of near 5 min for all the stimuli tested (Figure 2C). Extracellular ATP returned to basal levels at 10 min when mobilizing 12–37% of medium. When the stimulus was the mobilization of 50% of the culture medium, the basal levels of extracellular ATP were still not restored after 10 min (Figure 2C). The maximal increase in extracellular ATP was directly dependent on the stimulus magnitude (Figure 2D). For the next experiments, the pipetting of 37% of the culture medium was used.

Considering that undifferentiated RAW cells were still observed after RANKL incubation (Figure 2A), osteoclasts were then purified by a serum-gradient, after the incubation with RANKL (see Appendix A for osteoclast-purification characterization). All the following experiments were performed with these purified osteoclasts. Gradient-isolated osteoclasts showed no differences in resting levels of extracellular ATP than the non-purified ones (276 ± 27 and 215 ± 23 pmol ATP/mg protein, respectively; *p* = 0.15, Mann–Whitney test). In purified osteoclasts, the mechanical stimuli also evoked a rapid ATP release, maintaining the 5-min half-life time (Figure 2E). However, the maximal extracellular ATP released by mechanical stimulation (37% volume pipetted) was four-fold lower in purified osteoclasts (Figure 2D–F). This could be explained by considering that, before the serum-gradient purification, the cell culture maintains a significant number of non-differentiated RAW264.7 monocytes (as seen in Figure 2A), and it was demonstrated that mechanical stimulation in monocytes evokes a prominent ATP release, reaching maximal extracellular ATP values of 6296 ± 1007 pmol ATP/mg protein with the 37% stimulus (Appendix A).

As a control to rule out the possible effect of cell lysis during the mechanical stimulation of cells, lactate dehydrogenase (LDH) was addressed in all the types of experiments, and no increase in extracellular LDH was detected up to 10 min (Figure 2G and Appendix A).

Considering that P2X_7_ receptor (P2X_7_R) has been previously suggested as a conduit for ATP release in mouse-derived osteoclasts at rest [32], the effect of its selective and non-competitive antagonist Brilliant Blue G (BBG) was addressed. Interestingly, the pre-incubation of RAW-derived osteoclasts with 50 nM BBG abolished the ATP release evoked by mechanical stimulation (Figure 2F). The selective blockade of Pannexin-1 hemichannels with 5 µM carbenoxolone did not significantly reduce ATP release after mechanical stimulation (Figure 2F). Pre-incubation of cells with 100 µM glibenclamide, a general blocker of ABC transporters and exocytic ATP release, did not alter the increase in extracellular ATP levels evoked by mechanical stimulation (Figure 2F). It was also demonstrated the expression of P2X_7_R mRNA in osteoclasts by qRT-PCR (see the expression pattern of P2Y/P2X receptors in RAW 264.7 cells in Appendix A), and the protein expression by immunofluorescence (Figure 2H).

All these data reinforce that RAW-derived osteoclasts release ATP by mechanical stimulation in a regulated and non-lytic manner.

### 2.3. Exogenous ATP Promotes Protein Synthesis in Mouse Isolated FDB Muscle, through the P2R-Akt-mTOR Pathway

A classical signaling pathway for promoting protein synthesis in skeletal muscles is the Akt-mTOR [45,46]. We have previously demonstrated that exogenous ATP increases Akt phosphorylation in cultured myotubes, which mediates GLUT4 translocation and glucose uptake [41]. In the present work, it was addressed if exogenous ATP activates the Akt-mTOR pathway leading to an increase in protein synthesis in the mouse FDB muscle in vitro. The whole isolated muscle was used, without enzymatic dissociation of its fibers, to resemble the physiological environment that has to surpasses a bone-to-muscle signaling molecule.

Firstly evaluated was the metabolization rate of the 100 µM ATP when incubating with the whole FDB muscle in vitro. The ATP concentration was moderately reduced by 30% in the first 15 min, then stabilized and reached a 40% reduction at 90 min (Appendix A). Then, it can be concluded that exogenous ATP remains at high concentration for prolonged periods of time in this model.

Exogenous 100 µM-ATP increased in a time-dependent manner the phosphorylation of Akt (S473) and the ribosomal protein S6 (S235/236) in FDB muscles. While Akt phosphorylation was significantly increased from 7 min, the increase in S6 phosphorylation was only significant at 20 min (Figure 3A,B). ATP induced Akt phosphorylation in a concentration-dependent manner, from 1 µM, with a maximal response at 3 µM (Figure 3C). Then, 3 µM ATP was used in the next experiments for evaluating the phosphorylation of downstream proteins of the pathway. ATP increased 1.8-fold the phosphorylation of mTOR (S2448) and p70S6K (T389), after 20 min incubation (Figure 3D,E). The phosphorylation of 4E-BP1 (T37/46) increased 1.5-fold after 10 min stimulation (Figure 3F).

It was then evaluated whether exogenous ATP, in addition to activating the Akt-mTOR pathway, induced protein synthesis in isolated FDB muscle. Exogenous 3 µM ATP evoked a 1.7-fold increase in puromycin incorporation to newly synthesized proteins in FDB muscle, indicating an increase in global protein synthesis. That was completely reversed when ATP was co-incubated with 100 µM suramin, a general antagonist for all P2Y and P2X receptor subtypes. Interestingly, FDB incubation with 3 µM UTP, a P2Y_2_-P2Y_4_ selective agonist, increased by two-fold the puromycin incorporation (Figure 4A). In order to address the dependence of ATP-evoked protein synthesis on the Akt-mTOR signaling pathway in FDB muscle, coincubation with selective blockers was tested. Either the blockade of upstream PI3K with 50 µM LY294002 or the mTOR inhibitor Rapamycin (100 nM) abolished the increase of puromycin incorporation evoked by 3 µM ATP (Figure 4B).

In sum, these results suggest that extracellular ATP increases the protein synthesis in the whole FDB muscle through a P2Y/Akt/mTOR signaling pathway.

### 2.4. Mechanically Stimulated Osteoclasts Lead to Protein Synthesis in Co-Cultured FDB Muscle, through ATP Release and Activation of the P2R-Akt-mTOR Pathway

To demonstrate if released ATP was the signaling molecule that linked mechanical stimulation of osteoclasts with increased protein synthesis in whole FDB muscle in vitro, it was returned to the co-culture protocols in Transwell^®^ chambers. First of all, it was evaluated if extracellular ATP released by mechanically stimulated osteoclasts at the upper chamber could diffuse to the lower compartment through the 0.4 µm pore-size membrane. The lower chamber had just culture medium, without FDB muscle. In those conditions, mechanical stimulation of osteoclasts by moving 37% of culture medium at the upper compartment evoked a rapid and high increase in extracellular ATP at the lower one, reaching up to 2.5 µM concentration 15 s after the stimulus, with a half-life time of about 5–10 min (Figure 5A).

Mechanical stimulation of osteoclasts in the upper compartment increased 2.4-fold the puromycin incorporation in whole FDB muscles at the lower chamber (Figure 5B). This process was totally blocked with 2U/mL Apyrase, an ectonucleotidase that metabolizes ATP to AMP, as well as by using Suramin, the general antagonist of P2Y/P2X receptors (Figure 5B). Moreover, blockade of the Akt/mTOR pathway by using either LY294002 or Rapamycin also abolished the increase in global protein synthesis on FDB muscle evoked by mechanical stimulation of the osteoclasts (Figure 5B). These data confirm that mechanical stimulation of osteoclasts releases ATP, which evokes FDB global protein synthesis by activating a P2Y-Akt-mTOR signaling pathway.

## 3. Discussion

The concept of bone-muscle crosstalk through soluble molecules has developed substantially in the last decade. In the present work, we described for the first time that osteoclasts release ATP to the extracellular environment both basally and in response to mechanical stimulation, and that this molecule promotes the synthesis of proteins in isolated-FDB muscles through the Akt/mTOR pathway. We are considering here a physiological action of osteoclasts in response to bone remodeling, rather than a phenomenon of pathological resorption. That is, mechanical stimulation could induce bone remodeling and increase muscle protein synthesis to adapt to the load. Unlike most studies, which indicate that mechanical muscle traction promotes bone formation, it is possible that load sensing by bone cells promotes muscle mass increase. The results are summarized in a working model (Figure 6).

### 3.1. Mechanically Evoked ATP Release from Osteoclasts

One of the main findings of the present research is that purified osteoclasts release ATP to the extracellular medium after mechanical stimulation. In the literature, it has been considered that osteocytes are the cells responsible for mechanotransduction in bones, and there are recent studies that show that osteoblasts also respond to the movement of the environment [12]. It has been hypothesized that osteocytes and osteoblasts are the sources of extracellular ATP that would have a paracrine effect to stimulate osteoclasts [47,48]. In this work, we found that osteoclasts can also respond to mechanical stimuli by releasing ATP in a regulated way. These results, together with the evidence that mechanical stimulation evokes intracellular calcium oscillations in osteoclasts [49], as osteocytes and osteoblasts do, suggest that all remodeling cells in bone have mechanotransduction capacity. An interesting point to study in the future is whether these intracellular calcium oscillations mediated by the movement of the environment are produced by autocrine activation of P2Y/P2X receptors through released ATP. It would also be interesting to address whether the STIM1 and TRPV4 mechanosensitive calcium channels, which have been reported in the osteoclasts [49], are involved in the release of ATP post mechanical stimulation.

The release of ATP evoked by turbulent flow in osteoclasts was abolished with the pharmacological blockade of the P2X_7_R using BBG. However, the resting levels of extracellular ATP were not affected by P2X_7_R blockade, suggesting that a different pathway (either vesicular or conductive) could be involved in the basal ATP release. The latter partially differs from that reported by Brandao-Burch et al., which showed that basal ATP release in osteoclasts depends on both the P2X_7_R and a vesicular pathway [32]. This incongruency could be due to the cell type used (RAW264.7 vs. primary culture) since it has been demonstrated that the pharmacological blockade of the mammalian P2X_7_R varies even among strains of the same species [50].

Some previous reports suggest that BBG, used here as a P2X_7_R blocker, could also block the hemichannel Pannexin-1, another known conduit for ATP release. However, it requires concentrations in the 1–3 µM range [51,52]. Then, the concentration of BBG used in the present work (50 nM) ensures the selective blockade of the P2X_7_R. As an additional control, pre-incubation of cells with 5 µM carbenoxolone was used to selectively block Pannexin-1 hemichannels [53]. This treatment did not modify ATP release either at rest or after mechanical stimulation. So, the data suggest that Pannexin-1 hemichannels are not relevant for ATP release in our system.

It is a recurring question whether the P2X_7_R can serve as an ATP channel per se. P2X_7_R normally acts as a non-selective ionic channel for Na^+^/Ca^2+^, activated by extracellular ATP. However, some reports suggest that after P2X_7_R activation, molecules up to 900 kDa in size could permeate [54], which includes ATP. The release of ATP directly through a pore-forming P2X_7_R in astrocytes has been proposed, mentioning that P2X_7_R could vary conformation between a channel and a pore [55]. The same idea was proposed for constitutive ATP release in mouse osteoclasts [32].

Calcium entrance through P2X_7_R could promote ATP release either by exocytosis or by ABC transporters. To address these possibilities, it was tested pre-incubation with 100 µM glibenclamide, which is an ABC-transporter inhibitor [56]. Interestingly, glibenclamide also inhibits ATP storage into vesicles by inhibiting the VNUT (vesicular nucleotide transport), avoiding ATP loading into secretory vesicles, and then the release through exocytosis [57,58]. Glibenclamide did not modify the ATP release evoked by mechanical stimulation in purified osteoclasts. Then, participation of both exocytosis and several ABC-transporters in ATP release are excluded in our system. With this approach, we can not confirm that P2X_7_R is the direct pathway of ATP release in our system, but it is demonstrated that it is a relevant mediator for the process. Identifying the specific release pathway of ATP from the osteoclasts was not the main focus of the present work; then, future molecular and pharmacological experiments will be required to clarify this point.

The mechanical stimulation of cells occurs physiologically in bones following the movement of fluid through the lacunar-canalicular system [59,60]. The release of ATP from all the bone cells during loading cycles could be essential in maintaining bone structure and function, as well as signaling to neighbor tissues. Although we did not study here the autocrine effect of ATP released from purified osteoclasts, it will be interesting to test in future investigations whether the release of ATP after mechanical stimulation acts as an autocrine signal to foster differentiation and resorption activity in osteoclasts. It has been well described that osteoclasts express several P2Y/P2X receptor subtypes [61] that are involved in osteoclastogenesis [62].

### 3.2. eATP as a Protein Synthesis Inductor in Skeletal Muscle

The other relevant contribution of this work is the establishment of eATP as an inductor of protein synthesis in whole skeletal muscle, through activation of P2Y receptors and the Akt-mTOR signaling pathway. Previously it has been reported that skeletal muscle cells express multiple subtypes of P2X/P2Y purinergic receptors [39], which participate as mediators between plasma membrane depolarization and gene expression [39], promote the translocation of GLUT4 and glucose uptake [41], activate anti-apoptotic pathways [63] and regulate the production of ROS [64]. Recently it has been demonstrated that ATP administration led to muscle hypertrophy, or alleviates muscle atrophy, by activating the mTOR pathway [42]. In this work, they measured muscle hypertrophy by changes in total muscle mass, which could respond to either an increase in protein synthesis or a decrease in protein degradation. In the present work, it was directly demonstrated that eATP evokes protein synthesis in whole isolated muscle, measured by the increase of neosynthesized peptides. We demonstrated that the protein synthesis elicited by eATP is mediated by activation of the Akt-mTOR signaling pathway. Maximum Akt phosphorylation occurs with 3 µM ATP. This data suggests the participation of P2Y receptors in this process, because they are activated by ATP concentrations ranging from nM to low µM, while P2X-type receptors require ATP concentrations in the range of high µM to mM for activation [19]. Considering that in isolated fibers of mouse FDB muscle we have shown that the P2Y_2_ receptor is the most expressed subtype [65], it could be the intermediate receptor between extracellular ATP and Akt activation. As reinforcement to this point, and considering that the P2Y_2_ receptor has an equivalent sensitivity for ATP and UTP, here it was demonstrated that 3 µM UTP induces a 1.8-fold increase in protein synthesis in isolated FDB muscle. Interestingly, Ito et al. demonstrated that mTOR activation by eATP in C2C12 cells is dependent on P2Y_2_ receptor activation. However, contrary to our work, they showed that Akt phosphorylation is not a mediator between PI3K and mTOR activation. The latter suggests that trophic signaling pathways might be different in immature muscle cells, such as C2C12, than in mature skeletal fibers that are part of a structured, non-disaggregated tissue, such as freshly isolated whole skeletal muscle. [42].

Analyzing the possible downstream effects of Akt phosphorylation, the phosphorylation of mTOR (S2448), p70S6K (T389) and 4E-BP1 (T37/46) was studied after incubation with 3 µM ATP. It was demonstrated that mTOR and p70S6K increase their phosphorylation 1.8 times at 20 min, while the 4E-BP1 protein was phosphorylated 1.5 times at 10 min. This premature activation of 4E-BP1 has been reported in other studies, in which it was shown that apart from being activated by mTOR, it can be directly activated by Akt [66,67]. The latter would be a possibility, considering that Akt phosphorylation is detected as early as 7 min.

Considering that it has been described that the PI3K-Akt-mTOR pathway promotes the survival and growth of skeletal muscle [45,46], it was evaluated whether 3 µM ATP impacts the protein synthesis rate of isolated FDB muscle. Incubation of the complete muscle with 3 µM ATP increased puromycin incorporation by 1.42-fold using the SUnSET technique, which is an indicator of the level of neo-synthesized proteins. This effect was nullified when the muscle was pre-incubated with the pharmacological blockers suramin, LY294002 and Rapamycin, which suggests that the increase in the rate of protein synthesis induced by exogenous ATP is dependent on the P2-PI3KmTOR pathway, as it has been described in fibroblasts [43]. Interestingly, we have previously described that P2Y_2_R and PI3K are both participants of a multiprotein complex assembled at the T-tubules, involved in ATP signaling in skeletal muscle [68]. The latter reinforces the participation of these proteins in the signaling pathway that allows mTOR activation by exogenous ATP.

The role of exogenous ATP as an inducer of protein synthesis suggests that it could be a molecule responsible for trophically maintaining basal muscle mass, or for promoting muscle hypertrophy in pathophysiological conditions. Considering that ATP is a ubiquitous molecule, released by multiple cellular components of the musculoskeletal system (muscle fibers, bone cells, blood vessel cells, connective tissue fibroblasts) [18], it could participate in the fine regulation of muscle mass. This evidence opens a new area of clinical interest in pathologies such as muscular dystrophies, in conditions such as the loss of muscle mass during aging (sarcopenia), or in adaptive processes related to muscle use/disuse. Apart from its clinical relevance, the processes that control muscle mass are an attractive target in the area of exercise physiology and high-performance training.

### 3.3. ATP as a Signaling Molecule for Bone-Muscle Crosstalk

Strong associations between muscle and adjacent bone size have been reported throughout life, in which a larger muscle size is consistently related to a larger periosteal circumference [3]. For many years this positive relationship was justified by mechanical communication between muscle and bone, in which the muscle through contraction generates mechanical stimulation on the bone, regulating its mass [2,3,4]. However, in recent years, the muscle has been shown to release different molecules (called myokines), such as myostatin, irisin and interleukins that regulate bone physiology. In parallel, bone cells release molecules (called osteokines) capable of regulating homeostasis of muscle fibers, such as osteocalcin, prostaglandin E2 and Wnt-3a. This has led to the establishment of the concept of biochemical communication or crosstalk between the two structures [2,4,69,70,71].

The final significant contribution of this work is the positioning of ATP as a putative signaling molecule for bone-muscle crosstalk. Considering that ATP is a versatile molecule, which can be released from both bone and muscle cells and to have effects in both, we propose that it might be a bidirectional bone-muscle signal, acting either directly or indirectly. We consider that, if ATP is released from bone cells during loading to evoke bone remodeling and increase bone mass, it could also promote a rise in muscle mass, to allow the coordinated musculoskeletal adaptation. In the present work, we have shown that mechanical stimulation promotes ATP outflow from purified osteoclasts. In parallel, it was demonstrated that exogenous ATP promotes protein synthesis in isolated FDB muscle. The following approach was to assess whether the mechanical stimulation of osteoclasts in a co-culture system in Transwell*^®^* chambers with skeletal muscle was able to promote protein synthesis through the released ATP. With this experimental approach, in addition to the pharmacological blockade of different points of the route, it was demonstrated that the mechanical stimulation of purified osteoclasts increases protein synthesis in the FDB muscle, through the release of ATP to the extracellular environment and activation of the signaling pathway P2-PI3K-mTOR. It is interesting to note that the use of apyrase, which degrades the ATP released into the extracellular environment, totally blocks the induction of protein synthesis in muscle induced by mechanical stimulation of osteoclasts. This places the ATP as a relevant mediator of this process, either directly or by inducing the release of another signaling molecule.

When ATP is postulated as a communication molecule between bone and muscle cells, the passage of ATP through the different anatomical barriers that occur from the bone to the arrival to the muscle fiber must be considered. In our model, the most critical lacking is the periosteum, since the muscular fasciae such as epimysium, perimysium and endomysium are indeed present when whole isolated muscle is used. The periosteum is a barrier approximately 60 µm thick in mice [72], which comprises collagen and elastin fibers. Its structure is soft and hyper-elastic, which gives the bone the mechanical ability to absorb more energy [73]. It has been determined that the periosteum is a semi-permeable tissue. Based on the studies of Lai et al., who estimate the rate of transfer of molecules through the periosteum depending on their molecular weights [72], ATP would have a penetration time close to 10–20 s. In the ATP release tests from osteoclasts by mechanical stimuli, where concentrations of ATP 1.75 ± 1.1 µM were found, the half-life of the released ATP was close to 10 min. In metabolization assays of 3–100 µM of exogenous ATP in the presence of isolated FDB muscle, the half-life of exogenous ATP was longer than 90 min. In a previous study in myotubes derived from neonatal rats, it was shown that ATP released by electrical stimulation (20 Hz, 270 pulses, 1 ms each) had a half-life close to 30 min [39]. All these data suggest that, although the rate of metabolism of ATP depends on the concentration present, and the amount and type of nucleotidases expressed in each tissue, its half-life in contact with cells/tissues range between 10 min and more than one hour, which would be long enough to allow its passage through the periosteum.

An important point to keep in mind is that, physiologically, during muscle contraction many cells close to skeletal muscle release ATP into the extracellular medium. This is the case of motor neurons [74], vascular endothelial cells [75] and even the skeletal muscle fibers themselves [39]. This suggests that muscle contraction, which generates lacunar-canalicular flow in the bone, would cause the release of extracellular ATP from multiple cell types of the musculoskeletal system, which would participate in the maintenance of the mass and homeostasis of muscles and bones. This could partly explain why, in the absence of muscle activity or mechanical loading at the bone, these pathways do not act, thus promoting an imbalance between protein synthesis and degradation in the muscle, accounting for disuse atrophy.

It should also be mentioned that one possibility is that the ATP released from osteoclasts is not just a direct mediator of the induction of protein synthesis in muscle cells, but also promotes the release of other factors from osteoclasts, in larger amounts, with greater stability, or more power. Furthermore, considering that other cell types near the bone-muscle unit can release ATP (fibroblasts, motoneurons, vascular cells, and even skeletal muscle itself), it could be possible that ATP signaling acts as a relay race, inducing an ATP-induced ATP release pathway until it reaches the final effector cell. In our work, we can confirm that the release of ATP from cultured osteoclasts is capable of inducing protein synthesis in skeletal muscle in vitro. However, we cannot actually say that this occurs directly in vivo, and it is a very complex point to resolve. Advanced imaging approaches in vivo could help unravel this issue in the future.

The present work places extracellular ATP as a putative signaling molecule in the bone-muscle crosstalk.

## 4. Materials and Methods

All procedures involving animals were approved by the Institutional Animal Care and Use Committee of the Faculty of Dentistry of Universidad de Chile (Certificate N° 061501). The results are reported following the ARRIVE guidelines.

### 4.1. Cell Culture and Differentiation

The murine RAW 264.7 macrophage cell line from American Type Culture Collection (ATCC^®^TIB-71™) was cultured and maintained under standardized conditions (humidified atmosphere; 5% CO_2_; 37 °C) in Dulbecco’s modified Eagle’s medium (DMEM, Thermo Fisher Scientific, Waltham, MA, USA) supplemented with 10% fetal bovine serum (FBS), 100 U/mL penicillin, 100 μg/mL streptomycin, and 1 mM sodium pyruvate. Cells were split either when they reached about 80% confluence or every 2–3 d, using scraping and resuspension as previously described [76]. To induce monocyte differentiation to osteoclasts, the RAW 264.7 macrophages were cultured in the presence of recombinant human receptor activator of nuclear factor kappa B ligand (RANKL; 35 ng/mL; R&D Systems, Minneapolis, MN, USA) for 5–7 d [76,77].

### 4.2. Preparation of Osteoclast-like Cell-Enriched Populations

Even after incubation with 35 ng/mL RANKL for 5–7 d, RAW 264.7 cell cultures maintain a high proportion of undifferentiated monocytes. To enrich those cultures in osteoclasts, cells were harvested after the 5–7 d stimulation with RANKL and fractionated by FBS density gradient sedimentation, as previously described [76]. Cells were harvested, resuspended and layered on a discontinuous FBS gradient (15 mL of 40% FBS overlying 15 mL of 70% FBS, both in Moscona’s High Bicarbonate—MHB-Buffer). Tubes were undisturbed for 30 min at room temperature prior to fractions collection. The top fraction (17 mL), corresponding to a major population of non-differentiated macrophages/monocytes, and the bottom fraction (12 mL) containing the multinucleated osteoclasts, were collected separately from the FBS gradient and centrifuged. Cells were seeded in DMEM with 10% FBS and maintained overnight before TRAP staining, mechanical stimulation or harvesting for qRT-PCR analyses. The characterization of gradient fractions is detailed in Supplemental Material (Appendix A).

### 4.3. TRAP Staining

This procedure was carried out using the Acid Phosphatase Leukocyte (TRAP) Kit (#387A, Sigma-Aldrich Corp, St. Louis, MO, USA), according to the manufacturer’s instructions. Cells seeded on coverslips were washed with ice-cold phosphate-buffered saline buffer (PBS; 137 mM NaCl, 2.7 mM KCl, 10 mM Na_2_HPO_4_, 1.8 mM KH_2_PO_4_, pH 7.4) and fixed at room temperature for 30 s. Then, the cells were washed with distilled water and incubated with TRAP reagent at 37 °C for 60 min. TRAP reagent was removed with distilled water, and the nuclei were stained with hematoxylin for 2 s. Images were acquired in a light microscope (Motic BA310E) at 40X magnification and analyzed using the ImageJ software (NIH, Bethesda, MD, USA) [78]. Osteoclasts were defined as TRAP-positive cells with three or more nuclei. At least three images per cover were analyzed.

### 4.4. Total RNA Extraction, Reverse Transcription and Quantitative Real-Time PCR (qRT-PCR)

Total mRNA was obtained from cell cultures using Trizol™ reagent (Thermo Fisher Scientific, Waltham, MA, USA), according to the manufacturer’s instructions. cDNA was obtained from 2 µg of total RNA by using the High-Capacity cDNA Reverse Transcription Kit (#4368814, Applied Biosystems, Pleasanton, CA, USA), as indicated by the manufacturer’s protocol.

The qRT-PCR was carried out in the StepOne™ Real-Time PCR System (Thermo Fisher Scientific, Waltham, MA, USA) using the Brilliant III Ultra-Fast SYBR^®^ Green QPCR Master Mix (#600882, Agilent Technologies, Santa Clara, CA, USA). The sequences of the primers used to amplify the cDNA of osteoclast-markers, or P2X/P2Y receptors, are detailed in the Supplementary Material (Appendix A). All primers were standardized to render an efficiency between 95% and 105%. The thermocycling protocol included 95 °C for 20 s followed by 40 cycles of 95 °C for 3 s and 60 °C for 30 s. The amplification procedure was verified by melting curve analysis. The results were normalized to Glyceraldehyde-3-phosphate dehydrogenase (Gapdh) expression (housekeeping) and reported according to the 2^−ΔΔCT^ method [79].

### 4.5. Fluorescence-Activated Cell Sorting

In order to determine the relative proportion of monocytes and osteoclasts through the differentiation and purification steps, cells were analyzed by fluorescence-activated cell sorting (FACS). Cells were detached, washed, fixed in 100 mL of PBS-1% BSA-1% paraformaldehyde, and injected in the flow cytometer (FACScan™, BD Biosciences, CO, USA). In each condition, 30.000 cells were sorted according to the size (FSC-H) and internal complexity/granularity (SSC-H). Data from FACS were analyzed using a free flow cytometry data analysis software (Flowing Software 2.5.1, Turku Centre for Biotechnology, Turku, Finland).

### 4.6. Mechanical Stimulation and Extracellular ATP Measurement

To produce the mechanical stimulation on the cell cultures, the movement of the medium in 35-mm Petri dishes was carried out by pipetting, as previously described for cultured osteoblasts in monolayer [80]. The mechanical stimulus consisted of the displacement of 12%, 25%, 37%, or 50% of the culture-medium volume, every 1s, for ten times, leading to a turbulent fluid flow of 3.75 × 10^−^^4^, 7.5 × 10^−^^4^, 11.1 × 10^−^^4^ or 15 × 10^−^^4^ L/s, respectively. The periodicity of mechanical stimulation by pipetting was standardized using a digital metronome. The procedure was performed carefully, with the tip of the micropipette adhered to the wall of the culture plate in a 45° angle, avoiding direct disturbance of the cells that could lead to detachment and death. ATP levels in the extracellular medium were quantitated using the CellTiter-Glo^®^ Luminescent Cell Viability Assay (Promega, Madison, WI, USA). To determine the kinetics of ATP release, extracellular media aliquots from mechanically stimulated or undisturbed cells were removed at different times post-stimulation and measured. A standard curve of ATP (1 fmol-100 pmol) was performed each time to interpolate the sample values. Data were expressed as pmol eATP/mg protein at different times post mechanical stimulation or as the maximal increase of eATP after the stimulus relative to its control level. Cell viability was parallel measured in all the assays to discard ATP release due to cell damage. Both the mechanical stimulation and the collection of the medium for ATP measurement were always performed by the same experimenter (CM-J), allowing calibration and avoiding inter-individual errors

### 4.7. Cell Viability

The quantification of lactate dehydrogenase (LDH) activity in the extracellular medium was used to determine the cell viability after mechanical stimulation of the cells, using an LDH cytotoxicity detection kit (Thermo Fisher Scientific, Waltham, MA, USA). The absorbance of each sample was measured at 490 nm using a microplate reader (Synergy HTX Multi-Mode Reader, Biotek, VT, USA).

### 4.8. Immunofluorescence

Osteoclast-like cell-enriched populations seeded in coverslips were rinsed with PBS, fixed in ice-cold methanol, and permeabilized with PBS-0.1% Triton X-100. Then, the covers were rinsed with PBS and blocked with PBS-1% BSA for 1 h at room temperature. After that, cells were washed with PBS and incubated with an anti-P2X_7_R antibody (Santa Cruz Biotechnologies, TX, USA), at 4°C overnight. Then, cells were washed with PBS and incubated for 1 h with the secondary antibody Alexa Fluor-555 (Thermo Fisher Scientific, Waltham, MA, USA). For nuclei staining, cells were incubated with Hoechst (Thermo Fisher Scientific, Waltham, MA, USA) for 10 min. Samples were finally washed with PBS, mounted in Dako anti-fading reactive (Dako, Glostrup, Denmark) and stored at 4 °C until use.

### 4.9. Muscle Dissection and Stimulation

BALB/c mice (8 weeks old, 18–25 g) were obtained from the Experimental Platform of the Faculty of Dentistry (Universidad de Chile). Standard animal room conditions (48–50% humidity; 20 ± 2 °C; 12 h light/dark cycle), and ad libitum water and food (LabDiet^®^ JL Rat and Mouse/Auto 6F 5K67) were maintained. FDB muscles were isolated from BALB/c mice as previously described [81], and stabilized in an organ culture bath for 2 h in DMEM supplemented with 1 mM sodium pyruvate, 100 U/mL penicillin, 100 µg/mL streptomycin and 1% horse serum (HS), at 37 °C. After that, muscles were stimulated with exogenous ATP at selected concentrations (0.1–100 µM), for different times (3–30 min). When blockers or inhibitors were used, they were incubated 30 min before and during the stimulation with ATP.

### 4.10. Immunoblot

FDB muscles were processed with a rotor/stator tissue homogenizer (Biospec, Bartlesville, OK, USA) in 150 µL of ice-cold lysis buffer (20 mM Tris-HCl, 1% Triton X-100, 2 mM EDTA, 10 mM Na_3_VO_4_, 20 mM NaF, 10 mM sodium pyrophosphate, 150 mM NaCl, 1 mM PMSF, completeTM protease inhibitor cocktail, pH 7.4). The cell lysates were sonicated for 3 min, incubated on ice for 30 min, and centrifuged to remove debris. The protein concentration was determined by the turbidimetric assay with sulfosalicylic acid. Proteins resolution by SDS-PAGE (7–15%) and immunoblot were performed as previously detailed [38]. Protein staining was performed with the RapidStepTM enhanced chemiluminescence (ECL) reagent (EDM Millipore, Bedford, MA, USA). Images were acquired in an Amersham Imager 600 (GE Healthcare Life Sciences, Piscataway, NJ, USA) and densitometry was analyzed with the ImageJ Software (NIH, Bethesda, MD, USA).

### 4.11. Protein Synthesis Assay

The Surface Sensing of Translation (SUnSET) method was used for monitoring the global protein synthesis in the whole muscle. Puromycin, by structural analogy to aminoacyl-tRNAs, is incorporated into the nascent polypeptide and prevents the elongation; then, an anti-puromycin antibody is used to detect tagged proteins by immunoblot [82]. Dissected FDB muscles were stabilized for 2 h in DMEM 1% HS at 37 °C, as previously detailed. Then, the medium was replaced with a fresh one supplemented with 1 μM puromycin (EDM Millipore, Bedford, MA, USA), which was maintained for 45 min (pulse). The stimulation with exogenous ATP or mechanical perturbation started simultaneously with puromycin addition. When blockers or inhibitors were used, they were incubated 30 min before and during the puromycin. After the 45-min pulse, a 30-min chase was performed in DMEM 1% HS. Finally, muscles were processed for puromycin detection by immunoblot, as previously described. Densitometric analyses of immunoblot staining considered the entire lane. Either GAPDH signal or ponceau staining was used indistinctively for normalization.

### 4.12. Paracrine Communication Assay (Transwell^®^ Chambers)

The ability of purified osteoclasts to have paracrine communication with the FDB muscle was assessed by the Transwell^®^ assay. This test was performed on 12-well Transwell^®^ plates provided with a polyester membrane with 0.4 µm pores (Corning, New York, NY, USA). In the upper chamber, 75,000 purified osteoclasts were seeded; DMEM 10% FBS was added to the upper and lower chamber and cells were maintained for 24 h at 37 °C. Then, the isolated FDB muscle was placed in the lower chamber, and the culture medium was replaced with DMEM 1% HS, proper to protein synthesis assays. After 2h-stabilization, 1 μM puromycin was added to the upper and lower chambers. Immediately, mechanical stimulation to cultured osteoclasts was carried out in the top compartment. The protein synthesis detection in the FDB muscle was performed as detailed in the previous section. As control of an undesirable influence of mechanical stimulation directly on the FDB muscle, an assay was performed by mechanical stimulation of the upper chamber with no seeded osteoclasts, maintaining FDB muscle in the lower chamber.

### 4.13. Statistical Analysis

Data of n experiments were expressed as mean ± standard error of the mean (SEM). Non-parametric tests were used to evaluate significance. Mann–Whitney test was used for comparing a single condition with a control. For multiple comparisons, the Kruskal–Wallis test followed by the Bonferroni post hoc test was used. A *p* value < of 0.05 was considered statistically significant.

## 5. Conclusions


Purified osteoclasts release ATP to the extracellular medium after mechanical stimulation, in a regulated and non-lytic way.eATP is an inductor of protein synthesis in skeletal muscle, through activation of P2Y receptors and the Akt-mTOR signaling pathway.Mechanical stimulation of purified osteoclasts increases protein synthesis in a co-cultured FDB muscle, through the release of ATP to the extracellular environment and activation of the P2-PI3K-mTOR signaling pathway.Then, ATP is a possible signaling molecule for bone-muscle crosstalk. Considering that ATP is a ubiquitous molecule, released by multiple cellular components of the musculoskeletal system, it could participate in the fine regulation of muscle mass. This evidence opens a new area of clinical interest in muscle pathologies, in conditions such as the loss of muscle mass during aging (sarcopenia), or in adaptive processes related to muscle use/disuse.


## Figures and Tables

**Figure 1 ijms-23-09444-f001:**
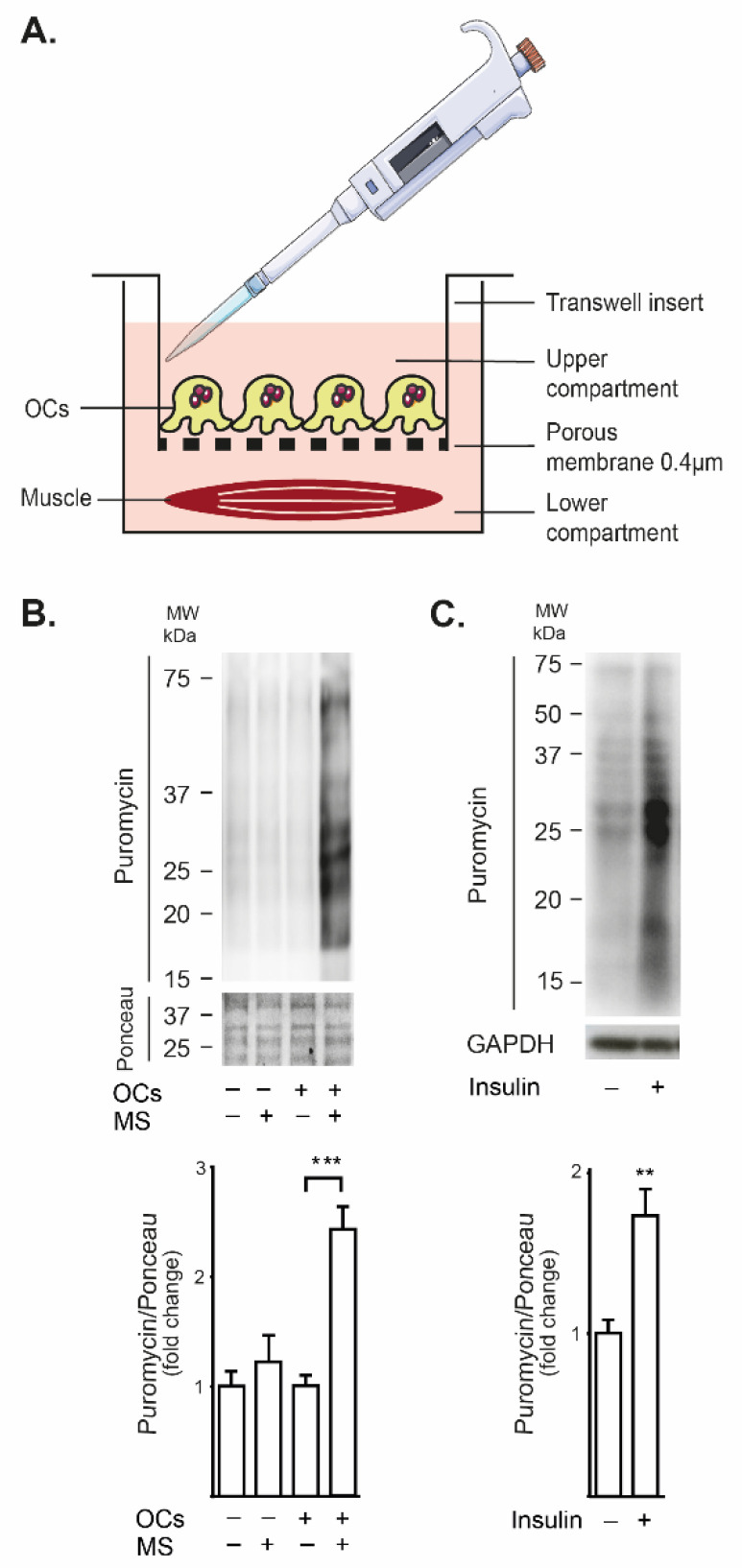
Mechanical stimulation of osteoclasts evoked protein synthesis in FDB muscle in vitro, through secreted mediators. (**A**) Schematic view of the Transwell^®^ co-culture system. Osteoclasts (OCs, derived from RAW264.7 cells, differentiated with RANKL and purified by serum gradient) were seeded in the upper chamber, and the isolated FDB muscle was placed in the lower compartment. Osteoclasts were mechanically stimulated by gently pipetting up and down (37% of the extracellular medium, ten times) in the upper compartment. Protein synthesis in the FDB muscle was estimated by puromycin incorporation (SUnSET method, 45 min pulse, 30 min chase). (**B**) Mechanical stimulation (MS+, lanes 1, 4) at the upper chamber evoked protein synthesis in FDB muscle only when osteoclasts were seeded (OCs+, lanes 3, 4). Puromycin immunoblot (top) and its quantification (bottom) are shown. (**C**) Incubation with 100 nM Insulin was used as a positive control of protein synthesis induction in FDB muscle. Mean ± SEM of three independent experiments is graphed. ** *p* < 0.01, *** *p* < 0.001, Mann–Whitney test, comparing MS− vs. MS+ in each case (**B**) or comparing without and with insulin (**C**).

**Figure 2 ijms-23-09444-f002:**
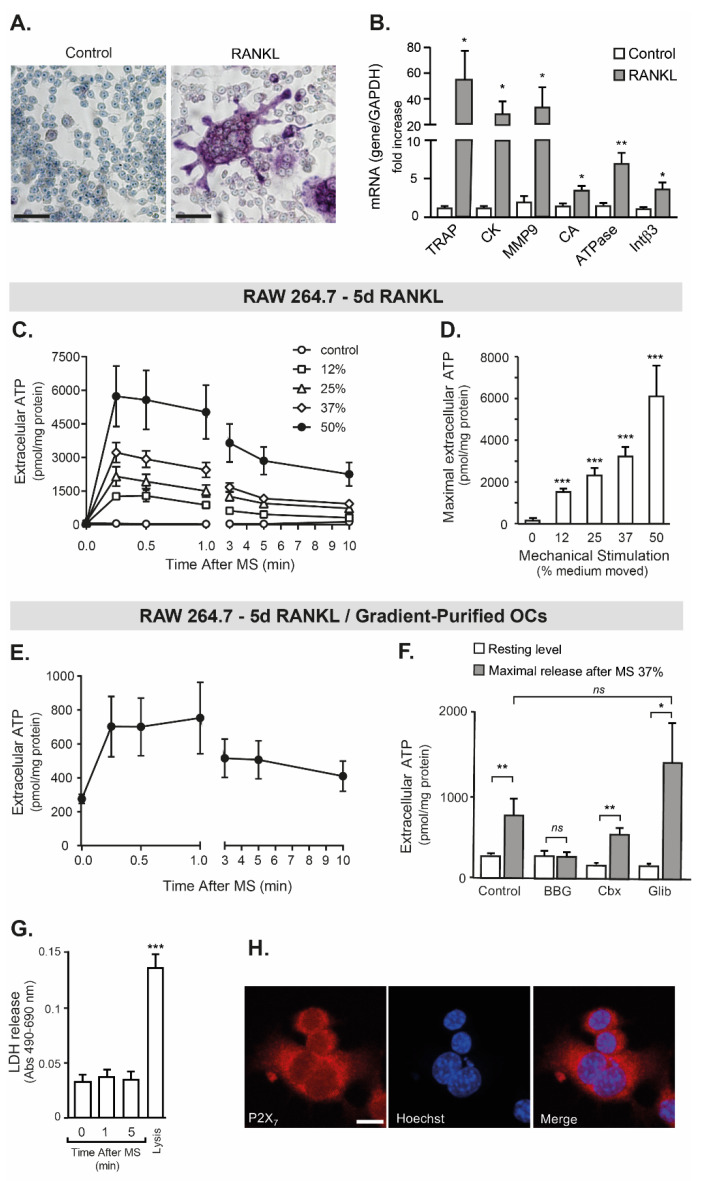
Mechanical stimulation induced ATP release from osteoclasts, dependent on the P2X_7_ receptor. (**A**) Incubation with RANKL evoked differentiation of RAW264.7 monocytes into TRAP+ multinucleated osteoclasts. TRAP expression was assessed by cytochemistry in RAW264.7 cells either in control (left panel) or incubated for 5d with 35 ng/mL RANKL (right panel). Cells were co-stained with hematoxylin. A representative image of *n* = 3 is shown. Scale bar: 25 µm. (**B**) RANKL (5d, 35 ng/mL) evoked the expression of osteoclastogenic markers in RAW264.7 cells. TRAP: Tartrate resistant acid phosphatase, CK: Cathepsin K, MMP9: Metalloprotease 9, CA: Carbonic anhydrase, ATPase: Lysosomal ATPase, Intß3: Integrin ß3. mRNA relative quantitation was performed using the ΔΔCT method, with Gapdh as the housekeeping gene. (**C**) Mechanical stimulation evoked ATP release from RAW 264.7 cells differentiated with RANKL. Cells were mechanically stimulated by pipetting different volumes of the extracellular medium (12, 25, 37, 50% of the total volume; one pipetting per second, ten times). The control condition corresponds to unperturbed cells. At different times post-stimulation, extracellular ATP was measured in the cell media. (**D**) Maximal ATP release depends on the magnitude of mechanical stimulation. Maximum values for different extent of mechanical stimulation tested are reported. (**E**–**G**) In osteoclasts-enriched fractions obtained from serum-gradient fractionation, the mechanical stimulation also evoked the ATP release in a controlled and non-lytic manner. Mechanical stimulation (mobilization of 37% total volume) evoked ATP release from purified osteoclasts; the kinetic is shown in (**E**), and the maximal release is quantitated in (**F**). The release of ATP evoked by mechanical stimulation was prevented by 30-min pre-incubation with 50 nM BBG (non-competitive antagonist of the P2X_7_R), but not with 5 µM carbenoxolone (Cbx, blocker of Pannexin-1 hemichannels) or 100 µM glibenclamide (Glib, blocker of ATP exocytosis and ABC transporters) (**F**). Extracellular levels of LDH were measured in all the assays to discard cell lysis promoted by mechanical stimulation (**G**). (**H**) RAW-derived osteoclasts expressed P2X_7_R. Immunofluorescence was performed in three independent experiments; nuclei were stained with Hoechst. Scale bar: 10 µM. * *p* < 0.05; ** *p* < 0.01, *** *p* < 0.001.

**Figure 3 ijms-23-09444-f003:**
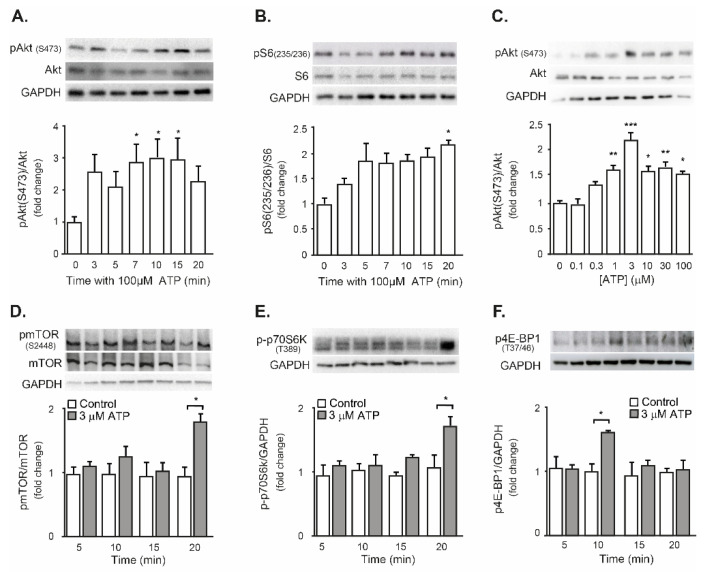
Exogenous ATP activated the Akt-mTOR transduction pathway in mouse FDB muscles. Kinetics of Akt phosphorylation (**A**) and S6 phosphorylation (**B**) evoked by ATP. Isolated FDB muscles were incubated with 100 µM ATP for 0–20 min. (**C**) Exogenous ATP evoked a concentration-dependent increase in Akt phosphorylation. FDB muscles were incubated for 10 min with 0–100 µM ATP. (**D**–**F**) 3 µM ATP evoked phosphorylation of mTOR (**D**), p70S6K (**E**) and 4E-BP1 (**F**) in isolated FDB muscles. Proteins from whole muscle extracts were resolved by SDS/PAGE (upper panels in each figure). Graphs in the bottom panels correspond to densitometric quantitation of 3–4 independent experiments (mean ± SEM), expressed as relative protein levels of phosphorylated proteins to the non-phosphorylated form (**A**–**D**) or GAPDH (**E**,**F**). * *p* < 0.05; ** *p* < 0.01, *** *p* < 0.001. Kruskal–Wallis test followed by Bonferroni post hoc, comparing each stimulation against the 0 (**A**–**C**). Mann–Whitney test, control vs ATP for each time (**D**–**F**).

**Figure 4 ijms-23-09444-f004:**
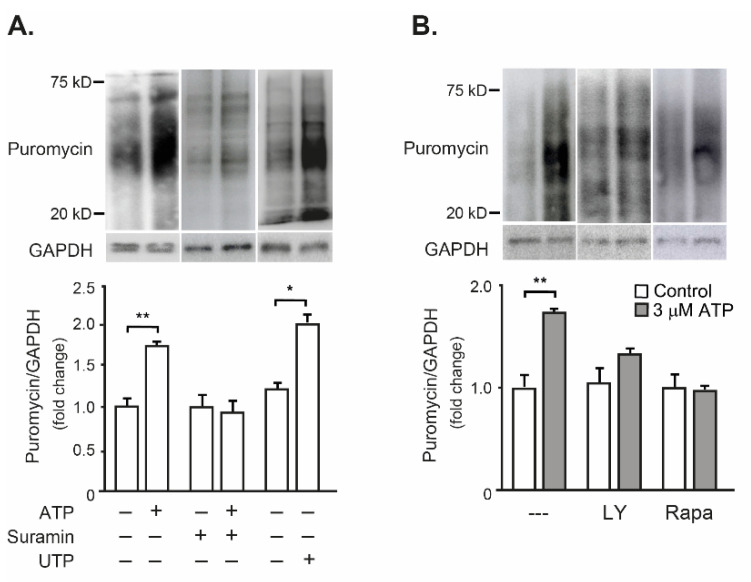
Exogenous ATP promoted protein synthesis in mouse FDB muscle, dependent on the P2YR-Akt-mTOR signaling pathway. Isolated FDB muscles were incubated without (−) or with (+) 3 µM ATP or UTP. A pulse of puromycin was performed during the incubation with the nucleotides. After that, 30 min chase with an unlabeled medium was carried out. Proteins from whole muscle extracts were resolved by SDS/PAGE, and neo-synthesized peptides were detected by immunoblot with an anti-puromycin antibody. For each figure, a representative immunoblot was shown at the top panel, and the quantitation of four independent experiments was graphed in the bottom panel (mean ± SEM). (**A**) ATP evoked protein synthesis in isolated FDB muscle, which is prevented by 30-min pre-incubation with 100 µM Suramin (P2Y/P2X receptor antagonist). UTP, a P2Y_2_-P2Y_4_ receptor agonist, also increased protein synthesis in isolated FDB muscle. (**B**) The increase in protein synthesis evoked by extracellular ATP in FDB muscle was abolished by 30-min pre-incubation with 50 µM LY294002 (LY, PI3-kinase inhibitor) or 100 nM Rapamycin (Rapa, mTOR inhibitor). * *p* < 0.05; ** *p* < 0.01, Mann–Whitney test.

**Figure 5 ijms-23-09444-f005:**
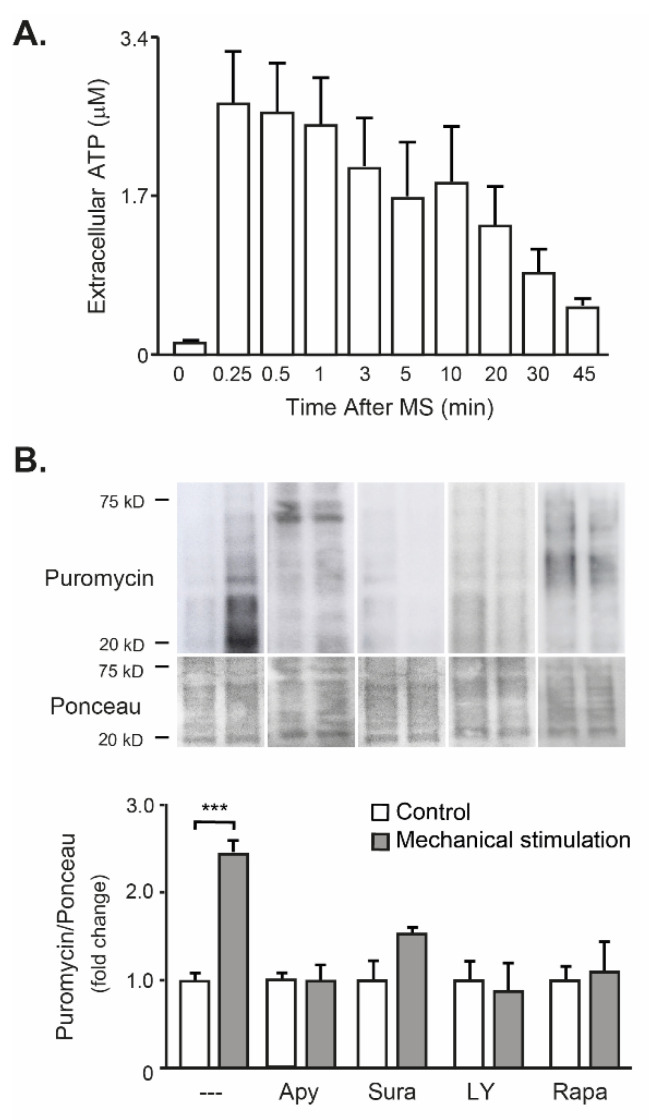
Extracellular ATP is the mediator released by the mechanical stimulation of osteoclasts that leads to protein synthesis in FDB muscle when they are co-cultured. The experimental model of Transwell*^®^* chambers was used, as detailed in Figure 1. (**A**) Measuring of eATP in the lower chamber, after mechanical stimulation of osteoclasts cultured at the upper compartment. This experiment was performed without any muscle in the lower compartment. Extracellular ATP was measured in the cell media (*n* = 3). (**B**) In an indirect co-culture system, with osteoclasts in the upper chamber and isolated FDB in the lower chamber, mechanical stimulation of osteoclasts leads to protein synthesis in FDB muscles. The latter was prevented by 30-min pre-incubation with 2 U/mL Apyrase (Apy, an ectonucleotidase that metabolizes extracellular ATP), 100 µM Suramin (Sura, P2Y/P2X receptor antagonist), 50 µM LY294002 (LY, PI3-kinase inhibitor) or 100 nM Rapamycin (Rapa, mTOR inhibitor). Osteoclasts were mechanically stimulated by pipetting 37% of the extracellular medium at the upper compartment, ten times. Protein synthesis was addressed at the FDB muscle by puromycin incorporation. Puromycin immunoblot (top) and its quantification (bottom) are shown. Mean ± SEM of 3–4 independent experiments is graphed. *** *p* < 0.001, Mann–Whitney test comparing control (undisturbed) with mechanical stimulation.

**Figure 6 ijms-23-09444-f006:**
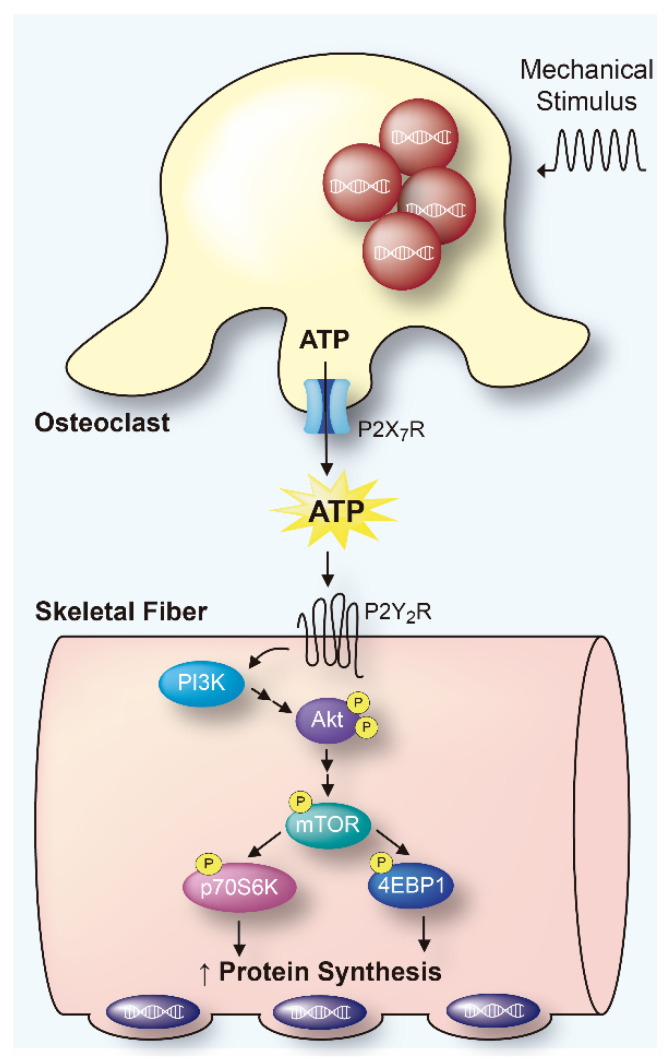
Graphic summary: ATP released by mechanically stimulated osteoclasts leads to protein synthesis in skeletal muscle through the Akt-mTOR signaling pathway.

## Data Availability

Not applicable.

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
