# Peer review of "Mechanical Disturbance of Osteoclasts Induces ATP Release That Leads to Protein Synthesis in Skeletal Muscle through an Akt-mTOR Signaling Pathway"

_ijms, 2022, doi:10.3390/ijms23169444_

Round 1

Reviewer 1 Report

The challenge for the authors is in how documents vary in their levels of comprehensiveness and detail, and the validity of the present manuscript depends on its methodological soundness. I have no negative considerations. The discussion is excellently written, as is each section. The study shines for precision, accuracy and novelty.

Author Response

We really appreciate the comments of reviewer 1. We now improved the text in order to consider all the reviewers' suggestions.

Reviewer 2 Report

Comments;

Here, the authors are suggesting that mechanical stimulation to osteoclasts release ATP, leading to protein synthesis in FDB muscle via activating the P2-PI3K-Akt-mTOR axis. They showed that osteoclasts cocultured in Transwell system released ATP to the extracellular medium in response to mechanical stimulation, which let the skeletal muscle induce protein synthesis. They prepared osteoclast-like cell from RANKL-stimulated RAW 264.7 cells to start with co-culture system.  Methods seem to be smart and results may be progressive and informative on the muscle-bone physiology, but some aspects should be considered again.

Followed are the points

In Fig 1, please mention the reason why chose the RAW cell as osteoclasts.

Please how a strong or constant stimulation was made to the osteoclast cells

Authors need to show the protein levels of the osteoclastogenic markers.

And provide the physical characteristics of the FDB muscle regarding the thickness, length, width and volume, etc.

In Fig 2C, it is wondering how constant periods for pipetting time and measuring ATP.  Please provide a precise method or a reference regarding Fig 2 C.

In Supplementary Figure 3, Kinetics of exogenous-ATP metabolization in myotubes were provided, please provide ATP consumption in the myocytes such as C2C12 cells, which increase the data significance

In Fig 3 A, B, and C, please calculate protein phosphorylations of the the Akt-mTOR pathway again after normalize with GAPDH intensities.

In Fig 4 or Fig 5B, please explain or discuss how long expose the muscle to antagonists or protein synthesis blockers. Are these same conditions as shown in Fig 3 or Fig 2?

Minor;

SuNSET à SUnSET in lane 524

Reviewer 3 Report

This study improve the knowledge on the field. The subject of the work is of interest and that the topic of the manuscript falls within the journal topic. Authors rationale is worthy of investigation. Although the manuscript is well written in each section I have some suggestions for the Authors which could improve it. 

Specific comments:

I suggest to avoid the use of personal form (i.e. our, we…) throughout the text.

The title well reflects the main aim and findings of the work.

Abstract is well written; Authors well summarize the purpose, results and significance of the study. However, to my opinion, the first paragraph of this section “Muscle and bone are tissues tightly integrated through mechanical and… skeletal muscle mass. Extracellular ATP is a signaling molecule released by- and acting on- bone and muscle cells.” Should be simplified. Moreover, Authors should indicate the statistical analysis applied on obtained data.

I suggest to make discussion more harmonic by deleting the subsections.

I suggest to add a conclusion section or paragraph in which Authors should summarize the main gathered findings and emphasize the significance of the study as well as propose new insights on this field according to the obtained findings.

Method section is well written; Authors describes the methodological approach clearly and meticulously.

Regarding statistical analysis, Authors wrote “Non-parametric tests were used to evaluate significance.” Did Authors perform a normality test to assess the normal distribution of data? Please clarify this aspect.

I would like to congratulate the authors for the figures which, in my opinion, are nice and well schematize the results gathered in the current study.
